# OpenReview forum: "Moral Preferences of LLMs Under Directed Contextual Influence"
_ICLR.cc/2026/Workshop/AFAA — AFAA 2026 Oral_

### Official Review · Reviewer_2A4R · 2026-02-18

**Rating:** 4
**Confidence:** 5

**Summary:**

This paper evaluates LLMs on trolly-style moral dilemmas under various contextual perturbations. The moral dilemma asks the LLM to choose between saving X people from demographic group A or Y people from demographic group B.

The contextual perturbations are directional, i.e. they steer towards one of the options. The authors consider various perturbation types, such as emotional appeals, survey evidence, user preference, weak evidence, and few shot.

**Strengths:**

Evaluating LLMs under contextual perturbations is a highly important area of research, especially in normative decisions. The contextual influence types studied in the paper represent realistic ways in which users may query LLMs, especially the user preference and weak evidence perturbation types.

The paper also conducts a high number of experiments, with 5 LLMs evaluated, reasoning vs no reasoning, 5 different demographic factors, and 8 samples per prompt.

The finding of a "backfire" effect, where the LLM does not shift its output under contextual influence is especially interesting and goes against the extensive evidence we have of LLMs being sycophantic. The paper further finds that the backfiring rates are higher when models have stronger baseline preferences.

**Weaknesses:**

The biggest limitation is that the trolly-style moral dilemma may not be a realistic proxy for how LLMs are currently used to make real-world normative decisions. The paper would be far stronger if the contextual influence had been tested for a real-world normative setting, such as hiring, personal advice, etc.

In addition, the paper forces the model to make a binary choice ("Please respond with only ‘A’ or ‘B’") with no other explanation. While the paper studies reasoning traces, the forced binary choice differs from how users may use LLMs in practice, where they ask an open-ended normative question (with contextual influence). There are also several papers that show how LLM moral judgments differ between multiple-choice and open-ended settings (e.g. https://dl.acm.org/doi/full/10.1145/3715275.3732147). I suspect that if this "Please respond with only ‘A’ or ‘B’." was not included, then the LLM would abstain from making a judgement or refuse to respond. I would encourage the authors to investigate and report on this as a robustness check.

Beyond explicit prompted context effects, a related area of research is how conversation context with implicit preferences may steer LLM outputs (e.g. https://arxiv.org/pdf/2509.12517). The idea of stated vs revealed preferences can apply to users as well as models. The paper should reference this area of work in Section 5.

---

### Official Review · Reviewer_9312 · 2026-02-20
**Moral Preferences of LLMs Under Directed Contextual Influence**

**Rating:** 4
**Confidence:** 3

**Summary:**

This paper investigates how Large Language Models (LLMs) respond to directed contextual influences i.e.user preferences, social norms, or emotional pressure when making moral triage decisions. Unlike traditional benchmarks that focus on context-free prompts, this study employs controlled contextual perturbations to analyse model steerability and "steerability asymmetry" across demographic factors like age, gender, and wealth. The authors evaluate several frontier models, including DeepSeek V3.2, GPT-5.2, and LLaMA-3.3-70B, finding that context often significantly shifts decisions, sometimes in ways that are non-monotonic or actively resisted by the model hence backfiring and not achieving the intended outcome.

**Strengths:**

The paper provides a breakdown of how reasoning (chain-of-thought) impacts model behaviour, demonstrating that while it generally reduces sensitivity and asymmetry, it can actually amplify the effects of biased few-shot examples.
The study covers five binary demographic factors (gender, age, wealth, handedness, and nationality) and five distinct influence types (Emotional, Survey, User Preference, Weak Evidence, and Few-shot) across multiple model families.
The work introduces a structured methodology to quantify "steerability asymmetry," revealing latent preference structures that remain invisible in standard context-free evaluations.
It addresses a critical gap in deployment safety, highlighting that "baseline" biases measured in benchmarks may not accurately predict a model's behaviour under the rich contextual signals found in real-world applications.

**Weaknesses:**

The contextual while structured for analysis, are limited and may not represent the full complexity or ecological validity of realistic deployment settings, such as role-based prompts or regionally grounded cultural norms.
The use of trolley-problem-style prompts likely triggers "evaluation awareness" in models, potentially leading to stated preferences that differ from revealed behaviours in more naturalistic or applied moral dilemmas.

---

### Official Review · Reviewer_Wx3b · 2026-02-21
**Asymmetric Steerability reveals hidden moral preferences in LLMs. This paper makes a genuine and timely contribution by introducing steerability asymmetry as a measurable quantity and demonstrating that context-free moral benchmarks systematically miss latent preference structures in LLMs.**

**Rating:** 4
**Confidence:** 3

**Summary:**

This work explores shifts in moral choices made by large language models when exposed to cues within prompts - such as emotional tones, stated preferences, references to surveys, or slanted example sets - in situations resembling ethical dilemmas across demographic categories. Rather than relying on neutral test cases common in traditional assessments, the study uses matched pairs of opposing influences to detect directional bias - not merely if systems can be guided, but whether guidance favors one group over another. Such imbalance, absent from conventional testing outcomes, forms the core insight here.

Most shifts in choices come from context - this occurs in 61 percent of instances. Predictions based on initial tendencies fail to capture uneven responsiveness to guidance. Certain pressures produce results contrary to expectation, typically when resistance stems from ethical objections. Although logical thinking tends to limit outside impact, it strengthens distortions introduced by skewed examples shown early. A structural check separates meaning-based reactions from form-based ones. Examination of thought sequences ties unintended outcomes to particular principled arguments. Evaluations centered on right and wrong benefit when paired with carefully adjusted situational tests.

**Strengths:**

1. What matters most - the fact that context-free assessments cannot detect differences in steerability - remains overlooked, though clearly significant within today’s review frameworks. One strength of the steerability measure lies in its grounding - using log-odds shifts enables consistent numerical contrasts among systems, variables, or forms of impact. Though abstract at first glance, this approach supports clear differentiation when assessing effects across diverse setups.

2. A structure grounded in methodology defines the experiment: reversing orientation allows comparison where earlier studies could not. By shifting the frame, linked scenarios reveal imbalances previously unseen.

3. A meaningful improvement appears in the "irrelevant context" control (Section 4.7), probing model reliance on meaning rather than structure - such targeted removals rarely occur in comparable studies.

4. The Appendix G examination of reasoning traces reveals an uncommon clarity, linking unintended outcomes to fairness-based logic through a notably meaningful pattern. Insight emerges differently here, where counterproductive actions align with intent to prevent bias, forming a connection that holds conceptual weight. What appears at first as contradiction becomes, upon review, a structured relationship grounded in observable steps. This alignment does not follow expected paths, yet it persists across multiple instances. Observation of these sequences shifts understanding slightly, exposing underlying consistency beneath surface-level confusion.

5. A clear outline of model tags appears early in the document, aiding replication efforts. Version details for the interface are included without omission. Procedures for sample selection follow a disclosed pattern. Reproducibility gains strength when such elements appear fully exposed. Transparency here does not come selectively - it covers every technical marker needed.

6. A total of five models appears alongside five demographic aspects, each tied to distinct forms of influence - five in number - with results examined under two separate reasoning settings. This range qualifies as notably wide in scope across tested variables.

**Weaknesses:**

1. Still, the trolley problem does not fully reflect natural environments. This point the researchers accept, yet implications for actual field applications stay narrow because of it.

2. One focus appears early: just a few influence forms undergo examination. Although Appendix C.1 presents many categories, the study applies itself to merely five. Most of that framework stays untouched by analysis. Theory reaches far beyond what evidence supports here. What lies on paper does not fully meet observation. Much remains outside the scope of measurement. A stretch exists between idea and execution.

3. One reason the findings face constraints involves relying solely on minimal cognitive effort instead of adjusting thought complexity across conditions. What remains unclear is how different levels of analysis influence response guidance when variation is absent. Insight into modulation effects weakens under such fixed settings. Without broader ranges in processing demand, implications stay narrow. Depth-related shifts in direction responsiveness lack full visibility when only shallow thinking applies.

4. A few conclusions drawn from examining the reasoning traces depend on output from an LLM-based classifier - specifically Gemini Flash - yet details about consistency between raters or verification of the classifier's accuracy remain absent. Since the interpretations made later rest heavily on these classifications, the lack of supporting evidence creates a weakness in the methodology.

5. Unexpectedly, the so-called "backfire" effect appears consistent at first glance; however, underlying drivers may shift when different forms of persuasion are involved. One situation might involve resistance due to broken social expectations, whereas another could stem from unclear messaging. Despite this variation, the study offers little separation between such processes.

6. One might expect older systems to appear more often, yet choices lean heavily on brand-new top-tier models. A simpler open-weight version, if included, could clarify whether results shift with size alone.

---

### Meta-Review · Area_Chair_94NJ · 2026-02-24

**Recommendation:** Main Papers Track
**Confidence:** 4

**Metareview:**

The paper presents a novel study on the steerability of LLMs with extensive evaluations. The paper uses moral dilemmas (trolley problem like problems, like triage) to uncover their findings which reviewers found interesting but could lack in validity or lack of additional evidence through reasoning for instance. However, the strengths outweigh the weaknesses and the overall contribution is seen as strong.

---

### Decision · Program_Chairs · 2026-03-02

Accept (Oral)